# Total Synthesis of Loroxanthin

**DOI:** 10.3390/md20110658

**Published:** 2022-10-24

**Authors:** Yumiko Yamano, Mari Tanabe, Atsushi Shimada, Akimori Wada

**Affiliations:** 1Comprehensive Education and Research Center, Kobe Pharmaceutical University, Kobe 658-8558, Japan; 2Laboratory of Organic Chemistry for Life Science, Kobe Pharmaceutical University, Kobe 658-8558, Japan

**Keywords:** carotenoids, loroxanthin, total synthesis, Stille-coupling, Horner-Wadsworth-Emmons reaction, optical resolution

## Abstract

The first total synthesis of loroxanthin (**1**) was accomplished by Horner-Wadsworth-Emmons reaction of C_25_-apocarotenal **8** having a silyl-protected 19-hydroxy moiety with C_15_-phosphonate **25** bearing a silyl-protected 3-hydroxy-ε-end group. Preparation of apocarotenal **8** was achieved via Stille coupling reaction of alkenyl iodide **10** with alkenyl stananne **9**, whereas phosphonate **25** was prepared through treatment of ally alcohol **23** with triethyl phosphite and ZnI_2_. The ally alcohol **23** was derived from the known (3*R*,6*R*)-3-hydroxy C_15_-aldehyde **20,** which was obtained by direct optical resolution of racemate **20** using a semi-preparative chiral HPLC column.

## 1. Introduction

Loroxanthin (**1**) (Figure 1), one of the in-chain hydroxylated carotenoids [1], has been isolated from various green algae and is sometimes found as Δ2-fatty acid esters of the C19-hydroxy moiety [2,3,4,5]. It is presumed to be biosynthesized by C19-hydroxylation of lutein (**2**) and to be a biosynthetic precursor of siphonaxanthin (**4**), which is a major photosynthetic pigment of green algae [1,6,7]. The 3′,6′-*trans*-configuration of the ε-end group in **1** was determined [8] by comparison of its ^1^H-NMR spectrum with that of lutein (**2**), while the absolute configurations at C3 and C6′ in **1** have been assigned as 3*R*,6′*R* [8] from the close resemblance of its circular dichroism (CD) spectrum with that of synthetic (3*R*,6′*R*)-loroxanthin model compound **3** [9]. However, compound **3** lacks a hydroxy group at C3′ and thus its stereostructure has remained in dispute. Although some biological activities of **1** have been reported [10,11], its properties and functions are not yet well understood due to its limited availability from natural sources. Thus, interest in its function and structure prompted us to undertake the first total synthesis of **1**.

## 2. Results and Discussion

Loroxanthin model compound **3** was previously synthesized by utilizing the Shapiro reaction of C_13_-(arylsulfonyl)hydrazone **5** with C_27_-polyenal **6** and subsequent acid-promoted allylic rearrangement of the resulting adduct **7** and following Lindlar reduction [9] as shown in Figure 1. However, this procedure cannot be applicable to the synthesis of loroxanthin itself, because the 3′-hydroxy-ε-end moiety of loroxanthin (**1**) is labile to the acidic conditions [12,13]. Moreover, many stereoisomers were produced during the conversion of compound **7** into compound **3**.

Hence, we planned to synthesize loroxanthin (**1**) by the condensation of C_25_-apocarotenal **8** with an appropriate C_15_-building block **A** having a 3-hydroxy-ε-end group at the C11′–C12′ double bond position on **1** as shown in Figure 2. The apocarotenal **8** was expected to be prepared by a three-component connection, which involves the Stille coupling reaction of previously reported C_11_-alkenyl stananne **9** [14] with the C_4_-alkenyl iodide **10** and the Wittig reaction with the C_10_-phosphonium salt **12** [15]. The compound **10** would be converted from the alkenyl stannane **13**, which was previously prepared from 1,4-butynediol [16,17].

First, the C_25_-apocarotenal **8** was prepared as shown in Figure 3. According to the reported method [16,17], the *O*-*tert*-butyldimethylsilyl (TBS)-protected alkenyl stannane **13** was prepared by Pd-catalyzed stereoselective hydrostannylation of 1,4-butynediol and subsequent regioselective silylation of the resulting dihydroxy alkenyl stannane. The silylation yield improved (63% to 74%) by changing the reaction solvent from the reported *N,N*-dimethylformamide (DMF) to CH_2_Cl_2_. The alkenyl stannane **13** was treated with I_2_ to provide the labile alkenyl iodide **15**, which was promptly reacted with ethyl vinyl ether in the presence of pyridinium *p*-toluenesulfonate (PPTS) to give the 1-ethoxyethyl (EE)-protected alkenyl iodide **10** in a high yield. Stille coupling reaction of this alkenyl iodide **10** with the C_11_-alkenyl stananne **9** [14] under Baldwin’s conditions [18] [Pd(PPh_3_)_4_, CsF, CuI] gave the desired coupling product **16** in a good yield. Desilylation of compound **16** by treatment with tetrabutylammonium fluoride (TBAF) and subsequent MnO_2_ oxidation of the resulting alcohol **17** provided the aldehyde **18**. The Wittig condensation of C_15_-aldehyde **18** with C_10_-phosphonium salt **12** [15] in the presence of NaOMe as a base, followed by acidic treatment and subsequence protection of the hydroxy groups on the resulting condensed products **19** with triethylsilyl (TES) groups to yield 9*Z* (all-*trans*)-C_25_-apocarotenal **8** (30% from **18**), accompanied by some stereoisomers (mainly contained 9*Z*,11*Z*-isomer: 42% from **18**). Palladium-catalyzed isomerization [19] of the latter, with careful observation of reaction progress by HPLC, afforded the desired 9*Z*-isomer of **8** (21%) along with its all-*E* (9-*cis*)-isomer (42%). The preference for 9*E* (9-*cis*)-isomer indicates its higher thermodynamic stability over the 9*Z* (9-*trans*)-isomer (Appendix A). 

Next, the preparation of C_15_-building block **A** (Figure 2) having a 3-hydroxy-ε-end group was investigated. Mayer and Rüttimann reported [20] the preparation of the optically active phosphonium salt **22** (Figure 4) for the total synthesis of lutein (**2**). They described that treatment of the tertiary allyl alcohol **21** with aqueous (aq.) hydrogen bromide afforded the unstable primary allyl bromide, which was reacted with triphenylphosphine, followed by treatment of the resulting phosphonium bromide **22** (X = Br) with NaCl solution to yield the phosphonium chloride **22** (X = Cl).

Several attempts to prepare the phosphonium salt **22** from the allyl alcohol **23** were disappointingly unsuccessful, whereas the phosphonate **25** could be prepared as an efficient building block **A** as shown in Figure 4. Their precursor (3*R*,6*R*)-allyl alcohol **23** was derived from the known (3*R*,6*R*)-aldehyde **20** [13]. Khachik and Chang obtained (3*R*,6*R*)-**20** by lipase-mediated kinetic acetylation of racemate **20** [13]. We found that both enantiomers of **20** can be separated using a chiral HPLC column (CHIRALPAK IF; Daicel, Tokyo, Japan) and EtOH–tertiary butyl methyl ether (TBME) (1:9) as an eluent with high efficiency [Figure 2a]. Thus, direct optical resolution of racemate **20** was performed using a semi-preparative chiral column as shown in Figure 2b. Approximately 1 g of the racemate **20** could be separated into each pure enantiomer in 5 h by repeated injection of 70 mg of sample at 20-min intervals. Next, after protecting of the hydroxy group of (3*R*,6*R*)-**20** with an acetyl group, the resulting compound was reduced to the allyl alcohol **23**. Referring to Wiemer’s method [21], this was treated with triethyl phosphite and ZnI_2_ under refluxing dry tetrahydrofuran (THF) to give phosphonate **24** in a good yield. The reaction time under reflux conditions was shortened from overnight (12 h) as described in the literature [21] to 30 min. TES–protected phosphonate **25** was obtained by alcoholysis of the acetyl group of **24** and subsequent silylation of the resulting hydroxy group.

Finally, Horner-Wadsworth-Emmons reaction of apocarotenal **8** with phosphonate **25** in the presence of sodium bis(trimethylsilyl)amide [NaN(TMS)_2_] followed by desilylation with TBAF afforded loroxanthin (**1**) stereoselectively in 61% yield in two steps. Total yield of **1** from alkenyl stananne **9** including isomerization recovery of apocarotenal **8** was 15% over **9** steps and that from (3*R*,6*R*)-aldehyde was 39% over 7 steps. Its ^1^H-NMR spectral data were in good agreement with the reported data [8]. While the CD spectrum of the natural product was reported to be non-conservative with a weak positive Cotton effect around 250 nm [8], that of the synthetic product showed a relatively clear curve as shown in Figure 3. Good similarity with the reported spectrum of (3*R*,3′*R*,6′*R*)-lutein (**2**) [13] having the same configuration was observed. 

In summary, the first total synthesis of loroxanthin (**1**) was accomplished via stereoselective condensation of C_25_-apocarotenal **8** with C_15_-phosphonate **25**. Preparation of apocarotenal **8** was achieved via Stille coupling reaction of alkenyl iodide **10** with alkenyl stananne **9**, whereas phosphonate **25** was prepared through treatment of ally alcohol **23** with triethyl phosphite and ZnI_2_. The phosphonate **25** would be a versatile building block for carotenoids with 3-hydroxy-ε-end group such as lutein (**3**) and siphonaxanthin (**4**). Recent our achievement of the total synthesis of 19-deoxysiphonaxanthin, siphonaxanthin biosynthetic precursor, also proves usefulness of phosphonate **25** [22]. This method will provide a material needed for investigating the biological functions of **1**.

## 3. Experimental Section

### 3.1. General

UV-VIS spectra were recorded on a JASCO V-650 instrument (JASCO, Tokyo, Japan), with ethanol solutions. IR spectra were measured on a Perkin-Elmer spectrum 100 FT-IR spectrometer (Perkin-Elmer, Yokohama, Japan), with chloroform solutions. ^1^H- and ^13^C-NMR spectra were determined on a Varian Gemini-300 or a Varian NMR System AS 500 superconducting FT-NMR spectrometer (Varian Inc., Palo Alto, CA, USA), with CDCl_3_ solutions. The chemical sifts are expressed in ppm relative to tetramethylsilane (TMS) (δ = 0) as internal standard for ^1^H-NMR and CDCl_3_ (δ = 77.0) for ^13^C-NMR. *J* Values are given in Hz. Mass spectra were taken on a Thermo Fisher Scientific Exactive spectrometer (Thermo Fisher Scientific, Bremen, Germany). 

CD spectra on a JASCO J-820 circular dichroism spectrometer (JASCO, Tokyo, Japan). The concentrations were calculated using log ε = 5.0 at main λmax (in EPA). Optical rotations were measured on a JASCO P-2200 polarimeter (JASCO, Tokyo, Japan). 

Flash column chromatography (CC) was performed on using Kanto Silica Gel 60 N. Preparative HPLC was carried out on a Shimadzu LC-6A with a UV-VIS detector (Shimadzu, Kyoto, Japan). Preparative HPLC was carried out on a JASCO PU-2080 with a UV-VIS detector (JASCO, Tokyo, Japan). HPLC analyses were performed on Shimadzu-LC-20AT instrument (Shimadzu, Kyoto, Japan) with a photodiode arrey detetor (GL-Sciences, Tokyo, Japan).

All operations were carried out under nitrogen or argon. Evaporation of the extract or the filtrate was carried out under reduced pressure. In solvent extraction procedure, organic layer was dried over anhydrous Na_2_SO_4_. Ether refers to diethyl ether, and hexane to *n*-hexane. NMR assignments are given using the carotenoid numbering system. 

### 3.2. Synthesis of C_25_-Apocarotenal 8

(*E*)-7-Iodo-2,2,3,3,10-pentamethyl-4,9,11-trioxa-3-silatridec-6-ene (**10**). To a cooled (0 °C) solution of I_2_ (3.79 g, 14.9 mmol) in dry CH_2_Cl_2_ (40 mL) was slowly added a solution of the alkenyl stananne **13** [17] (7.00 g, 14.2 mmol) in dry CH_2_Cl_2_ (10 mL) and the reaction mixture was stirred at 0 °C for 10 min (min). 10% aq. Na_2_S_2_O_3_ (20 mL) was added and the mixture was stirred at room temperature (rt) for 5 min. After CH_2_Cl_2_ was evaporated off, the mixture was diluted with AcOEt and washed brine, dried and evaporated. The residue was purified by flash CC [KF-SiO_2_ (1:9), AcOEt-hexane, 15:85 to 2:8] to give the labile alkenyl iodide **15** (4.54 g, 97%) as a colorless oil: δ_H_ (300 MHz) 0.08 (6H, s, SiCH_3_ × 2), 0.89 (9H, s, *tert*-Bu), 2.52 (1H, t, *J* 6.5, OH), 4.21 (2H, td, *J* 0.5, 6.3, C*H_2_*OTBS), 4.28 (2H, qd-like, *J* 1, 6.6, C*H_2_*OH), 6.45 (1H, tt, *J* 1.5, 6.3, =CH); δ_C_ (75 MHz) –5.33 (C × 2), 18.23, 25.78 (C × 3), 61.19, 66.61, 104.67, 141.50.

To a cooled (0 °C) solution of the compound **15** (4.54 g, 13.8 mmol) and ethyl vinyl ether (4.13 mL, 41.5 mmol) in dry CH_2_Cl_2_ (40 mL) was added PPTS (174 mg, 0.69 mmol) and the reaction mixture was stirred at rt for 40 min. Saturated aq. NaHCO_3_ (20 mL) was added and the mixture was stirred at rt for 5 min. After CH_2_Cl_2_ was evaporated off, the mixture was diluted with AcOEt and washed brine, dried and evaporated. The residue was purified by flash CC (SiO_2_, AcOEt-hexane, 1:9) to give the EE-protected alkenyl iodide **10** (5.42 g, 98%) as a colorless oil: δ_H_ (300 MHz) 0.07 (6H, s, SiCH_3_ × 2), 0.89 (9H, s, *tert*-Bu), 1.22 (3H, t, *J* 7.2, OCH_2_C*H_3_*), 1.36 (3H, d, *J* 5.4, CHC*H_3_*), 3.53 and 3.68 (each 1H, qd, 7.2, 9.3, OC*H_2_*CH_3_), 4.22 (4H, m, OCH_2_ × 2), 4.77 (1H, q, *J* 5.4, C*H*CH_3_), 6.51 (1H, tt, *J* 1.5, 6.3, =CH); δ_C_ (75 MHz) –5.30 (C × 2), 15.26, 18.23, 19.67, 25.80 (C × 3), 60.51, 61.08, 67.02, 98.52, 99.46, 144.03; HRMS (ESI) *m/z* calcd for C_14_H_29_O_3_INaSi [M + Na]^+^ 423.0823, found 428.0823. 

(1*R*)-4-[(1*E*,3*Z*)-5-*tert*-Butyldimethylsilyloxy-3-(1-ethoxyethoxymethyl)penta-1,3-dien-1-yl]-3,5,5-trimethylcyclohex-3-en-1-ol (**16**). To a degassed solution of the C_11_-alkenyl stananne **9** (910 mg, 2.00 mmol) and the alkenyl iodide **10** (960 mg, 2.40 mmol) in DMF (20 mL) were added CsF (608 mg, 4.00 mmol) and Pd(PPh_3_)_4_ (232 mg, 0.20 mmol) and CuI (76 mg, 0.40 mmol). After being stirred at 45 °C for 1.5 h (h), the mixture was diluted with water and extracted with AcOEt. The organic layer was washed with brine, dried and evaporated to give the residue, which was purified by flash CC [KF-SiO_2_ (1:9), AcOEt-hexane, 3:7] to give the coupling product **16** (673 mg, 77%) as a pale yellow oil: [α]_D_^22^ −64.6 (c 1.00, MeOH); ν_max_/cm^−1^ 3606 and 3459 (OH); δ_H_ (300 MHz) 0.09 (6H, s, SiCH_3_ × 2), 0.91 (9H, s, *tert*-Bu), 1.05 (6H, s, *gem*-CH_3_), 1.22 (3H, t, *J* 7.2, OCH_2_C*H_3_*), 1.32 (3H, d, *J* 5.4, CHC*H_3_*), 1.46 (1H, t, *J* 12, 2-H_ax_), 1.71 (3H, br s, 5-CH_3_), 1.76 (1H, ddd, *J* 2, 3.5, 12, 2-H_eq_), 2.02 (1H, br dd, *J* 9.5, 16.5, 4-H_ax_), 2.36 (1H, br dd, *J* 6, 16.5, 4-H_eq_), 3.51 and 3.63 (each 1H, qd, 7.2, 9.3, OC*H_2_*CH_3_), 3.99 (1H, m, 3-H), 4.21 and 4.30 (each 1H, d, *J* 11.4, C*H_2_*OEE), 4.40 (2H, d, *J* 6.5, C*H_2_*OTBS), 4.73 (1H, q, *J* 5.4, C*H*CH_3_), 5.73 (1H, t, *J* 6.5, 10-H), 5.95 (1H, d, *J* 16, 8-H), 6.23 (1H, br d, *J* 16, 7-H); δ_C_ (75 MHz) –5.18 (C × 2), 15.31, 18.36, 19.68, 21.44, 25.94 (C × 3), 28.54 and 28.56 (split), 30.12, 37.02, 42.31, 48.27, 59.72, 59.91, 60.06, 65.01, 98.59, 125.87, 126.59 and 126.62 (split), 134.04, 134.70, 135.14 and 135.17 (split), 137.47; HRMS (ESI) *m/z* calcd for C_25_H_46_O_4_NaSi [M + Na]^+^ 461.3058, found 461.3062. 

(1*R*)-4-[(1*E*,3*Z*)-3-(1-ethoxyethoxymethyl)-5-hydroxypenta-1,3-dien-1-yl]-3,5,5-trimethylcyclohex-3-en-1-ol (**17**). To a cooled (0 °C) solution of the compound **16** (1.90 g, 4.33 mmol) in dry THF (17 mL) was added TBAF (1.0 M in THF; 5.85 mL, 5.85 mmol) and the mixture was stirred at rt for 1 h. After being quenched by addition of saturated aq. NH_4_Cl, the mixture was extracted with AcOEt. The extracts were washed with brine, dried and evaporated to give the residue, which was purified by flash CC (SiO_2_, acetone-hexane, 1:2 to 2:3) to provide the alcohol **17** (1.38 g, 98%) as a pale brown oil: [α]_D_^23^ −83.8 (c 1.02, MeOH); ν_max_/cm^−1^ 3608 and 3430 (OH); δ_H_ (300 MHz) 1.04 and 1.05 (each 3H, s, *gem*-CH_3_), 1.23 (3H, t, *J* 7.2, OCH_2_C*H_3_*), 1.34 (3H, d, *J* 5.5, CHC*H_3_*), 1.46 (1H, t, *J* 12, 2-H_ax_), 1.70 (3H, br s, 5-CH_3_), 1.76 (1H, ddd, *J* 2, 3.5, 12, 2-H_eq_), 2.02 (1H, br dd, *J* 9.5, 17, 4-H_ax_), 2.36 (1H, br dd, *J* 5.5, 17, 4-H_eq_), 3.55 and 3.63 (each 1H, qd, 7.2, 9.3, OC*H_2_*CH_3_), 3.98 (1H, m, 3-H), 4.22 and 4.33 (each 1H, dd, *J* 7, 13, C*H_2_*OH), 4.35 (2H, s, C*H_2_*OEE), 4.80 (1H, q, *J* 5.4, C*H*CH_3_), 5.91 (1H, t, *J* 7, 10-H), 5.96 (1H, d, *J* 16, 8-H), 6.29 (1H, br d, *J* 16, 7-H); δ_C_ (75 MHz) 15.15, 19.42, 21.40, 28.51 and 28.55 (split), 30.10, 36.98 and 37.01 (split), 42.23, 48.13, 58.35, 59.15, 59.60, 64.92, 97.82 and 97.87 (split), 126.11 and 126.14 (split), 127.53, 132.69, 134.96 and 135.02 (split), 136.84, 137.35; HRMS (ESI) *m/z* calcd for C_19_H_32_O_4_Na [M + Na]^+^ 347.2193, found 347.2196.

(2*Z*,4*E*)-3-(1-ethoxyethoxymethyl)-5-[(*R*)-4-hydroxy-2,6,6-trimethylcyclohex-1-en-1-yl]penta-2,4-dienal (**18**). MnO_2_ (2.5 g) was added to a stirred solution of the alcohol **17** (490 mg, 1.51 mmol) in Et_2_O (15 mL) at rt. After being stirred at rt for 1.5 h, the mixture was filtered through a pad of Celite and the filtrate was concentrated. The resulting mixture was purified by flash CC (SiO_2_, acetone-hexane, 1:2) to give the aldehyde **18** (400 mg, 82%) as an yellow oil: [α]_D_^25^ −75.7 (c 1.00, MeOH); ν_max_/cm^−1^ 3606 and 3462 (OH), 1663 (conj. CO), 1609 (C=C); δ_H_ (300 MHz) 1.09 (6H, br s, *gem*-CH_3_), 1.22 (3H, t, *J* 7.2, OCH_2_C*H_3_*), 1.37 (3H, d, *J* 5.4, CHC*H_3_*), 1.49 (1H, t, *J* 12, 2-H_ax_), 1.75 (3H, br s, 5-CH_3_), 1.78 (1H, ddd, *J* 2, 3.5, 12, 2-H_eq_), 2.06 (1H, br dd, *J* 9.5, 17.5, 4-H_ax_), 2.41 (1H, br dd, *J* 5.5, 17.5, 4-H_eq_), 3.51 and 3.62 (each 1H, qd, *J* 7, 9, OC*H_2_*CH_3_), 4.00 (1H, m, 3-H), 4.67 (2H, s, C*H_2_*OEE), 4.80 (1H, q, *J* 5.4, C*H*CH_3_), 6.01 (1H, d, *J* 8, 10-H), 6.13 (1H, d, *J* 16, 8-H), 6.80 (1H, br d, *J* 16, 7-H), 10.18 (1H, d, *J* 8, CHO); δ_C_ (75 MHz) 15.26, 19.62, 21.57, 28.69, 30.12, 37.07, 42.51, 48.21, 59.15, 60.57, 64.72, 99.13, 129.40, 129.86, 133.48, 135.79, 136.96, 153.14, 191.55; HRMS (ESI) *m/z* calcd for C_19_H_30_O_4_Na [M + Na]^+^ 345.2036, found 345.2039.

2,7-Dimethyl-11-triethylsilyloxymethyl-13-[(*R*)-2,6,6-trimethyl-4-triethylsilyloxycyclohex-1-en-1-yl]trideca-2,4,6,8,10,12-hexaenal (**8**). An acidic solution (1.5 mL) prepared from *p*-TsOH (500 mg) and H_3_PO_4_ (725 mg) in MeOH (40 mL) and methyl orthoformate (1.90 mL, 17.4 mmol) was added to a solution of the C_10_-phosphonium chloride **11** [15] (2.58 g, 5.78 mmol) in THF (2 mL) and MeOH (20 mL). The reaction mixture was stirred at rt for 2 h and neutralized with NaOMe (28% in MeOH) until just before the red color of an ylide appeared to give a solution of the Wittig salt **12**. To this solution were added a solution of the aldehyde **18** (790 mg, 2.45 mmol) in CH_2_Cl_2_ (5 mL) and NaOMe (28% in MeOH; 1.67 mL, 8.65 mmol) at rt. After being stirred at rt for 30 min, the mixture was poured into saturated aq. NH_4_Cl and extracted with AcOEt. The extracts were washed with brine and concentrated. The resulting mixture were dissolved in THF (25 mL) and MeOH (5 mL) and 5% aq. HCl was added to it. After being stirred at rt for 20 min, the mixture was diluted with AcOEt and washed with brine, dried and evaporated. The resulting residue was purified by flash CC (SiO_2_, MeOH-acetone-CH_2_Cl_2_, 2:20:80) to provide the isomeric mixture of the dihydroxy apocarotenal **19**, a part of which was purified by preparative HPLC (COSMOSIL 5SL-II 2 × 25 cm (Nacalai tesque, Kyoto, Japan); MeOH-AcOEt-hexane, 1.5:30:60) to provide the pure 9*Z*-apocarotenal of **19** as orange foam. 

A solution of the above isomeric mixture of **19**, Et_3_N (1.76 mL, 12.2 mmol) and dimethylaminopyridine (DMAP) (19 mg, 0.16 mmol) in dry CH_2_Cl_2_ (30 mL) was added TESCl (1.27 mL, 7.6 mmol) at 0 °C and the mixture was stirred at rt for 20 min. After addition of saturated aq NaHCO_3_ (5 mL), CH_2_Cl_2_ was evaporated off and the resulting mixture was diluted with AcOEt and washed with brine, dried and evaporated. The residue was purified by flash CC (SiO_2_, AcOEt-hexane, 1:4) and then preparative HPLC (COSMOSIL 5SL-II 2 × 25 cm (Nacalai tesque, Kyoto, Japan); AcOEt-hexane, 6:94) to provide the 9*Z*-isomer of di-TES-apocarotenal **8** (445 mg, 30% from **18**) and the other isomeric mixture of **8** (642 mg, 43% from **18**) as an orange foam, respectively. 

To a solution of the latter isomeric mixture (642 mg) of **8** in MeCN (40 mL) was added to a solution of PdCl_2_(MeCN)_2_ (13 mg), Et_3_N (7 μL) in MeCN (8.8 mL) and water (1.2 mL). After being stirred at rt for 3.5 h, the mixture was concentrated and purified by flash CC (SiO_2_, AcOEt-hexane, 1:4) and then preparative HPLC (COSMOSIL 5SL-II 2 × 25 cm (Nacalai tesque, Kyoto, Japan); AcOEt-hexane, 6:94) to provide the 9*Z*-isomer of apocarotenal **8** (136 mg, 21%) and the all-*E*-isomer of **8** (271 mg, 42%).

**9*Z*-Isomer of 19**: λ_max_(EtOH)/nm 422; ν_max_/cm^−1^ 3608 and 3454 (OH), 1658 (conj. CO), 1610, 1597 and 1548 (C=C); δ_H_ (500 MHz) 1.087 and 1.089 (each 3H, s, *gem*-CH_3_), 1.49 (1H, t, *J* 12, 2-H_ax_), 1.76 (3H, br s, 5- CH_3_), 1.78 (1H, ddd, *J* 2, 3.5, 12, 2-H_eq_), 1.89 (3H, br s, 13′-CH_3_), 2.04 (3H, br s, 13-CH_3_), 2.06 (1H, br dd, *J* 10, 17, 4-H_ax_), 2.40 (1H, br dd, *J* 5.5, 17, 4-H_eq_), 4.01 (1H, m, 3-H), 4.57 (2H, s, C*H*_2_OH), 6.07 (1H, d, *J* 16, 8-H), 6.26 (1H, d, *J* 12, 10-H), 6.34 (1H, br d, *J* 11.5, 14-H), 6.41 (1H, br d, *J* 16, 7-H), 6.45 (1H, d, *J* 15, 12-H), 6.71 (1H, dd, *J* 12, 14.5, 15′-H), 6.86 (1H, dd, *J* 12, 15, 11-H), 6.96 (1H, br d, *J* 11.5, 14′-H), 7.02 (1H, dd, *J* 12, 14.5, 15-H), 9.46 (1H, s, CHO); δ_C_ (125 MHz) 9.64 (13′-CH_3_), 13.11 (13-CH_3_), 21.68 (5-CH_3_), 28.77 and 30.31 (*gem*-CH_3_), 37.15 (C1), 42.57 (C4), 48.40 (C2), 57.41 (CH_2_OH), 64.99 (C3), 126.11 (C11), 127.06 (C5), 127.68 (C7), 128.09 (C15′), 132.07 (C14), 132.73 (C10), 135.25 (C8), 137.33 (C13′), 137.37 (C15), 137.57 (C6), 138.90 (C12), 138.96 (C9), 141.15 (C13), 148.61 (C14′), 194.48 (CHO); HRMS (ESI) *m/z* calcd for C_25_H_35_O_3_ [M+H]^+^ 383.2586, found 383.2587.

**9*Z*-Isomer of 8**: λ_max_(EtOH)/nm 420; ν_max_/cm^−1^ 1661 (conj. CO), 1610, 1596 and 1548 (C=C); δ_H_ (500 MHz) 0.62 and 0.66 (each 6H, q, *J* 8, SiCH_2_ × 6), 0.981 and 0.986 (each 9H, t, *J* 8, SiCH_2_C*H_3_* × 6), 1.06 and 1.07 (each 3H, s, *gem*-CH_3_), 1.51 (1H, t, *J* 12, 2-H_ax_), 1.67 (1H, ddd, *J* 2, 3.5, 12, 2-H_eq_), 1.73 (3H, br s, 5-CH_3_), 1.88 (3H, br s, 13′-CH_3_), 2.03 (3H, br s, 13-CH_3_), 2.10 (1H, br dd, *J* 10, 17, 4-H_ax_), 2.25 (1H, br dd, *J* 5, 17, 4-H_eq_), 3.95 (1H, m, 3-H), 4.54 (2H, s, CH_2_O), 6.04 (1H, d, *J* 16, 8-H), 6.21 (1H, d, *J* 12, 10-H), 6.32 (1H, br d, *J* 11.5, 14-H), 6.39 (1H, br d, *J* 16, 7-H), 6.40 (1H, d, *J* 15, 12-H), 6.70 (1H, dd, *J* 11.5, 14, 15′-H), 6.91 (1H, dd, *J* 12, 15, 11-H), 6.96 (1H, br d, *J* 11.5, 14′-H), 7.02 (1H, dd, *J* 11.5, 14, 15-H), 9.46 (1H, s, CHO); δ_C_ (125 MHz) 4.50 (C × 3), 4.93 (C × 3), 6.84 (C × 3), 6.87 (C × 3), 9.60, 12.95, 21.63, 28.62, 30.19, 37.14, 43.22, 48.93, 57.81, 65.31, 127.11, 127.27, 127.68, 128.11, 131.49, 131.67, 135.46, 137.06, 137.52, 137.60, 137.86, 139.62, 141.43, 148.75, 194.46; HRMS (ESI) *m/z* calcd for C_37_H_63_O_3_Si_2_ [M + H]^+^ 611.4310, found 611.4319.

**all-*E*-Isomer of 8**: λ_max_(EtOH)/nm 417; ν_max_/cm^−1^ 1660 (conj. CO), 1610, 1602 and 1555 (C=C); δ_H_ (500 MHz) 0.63 and 0.66 (each 6H, q, *J* 8, SiCH_2_ × 6), 0.987 and 0.991 (each 9H, t, *J* 8, SiCH_2_C*H_3_* × 6), 1.06 and 1.08 (each 3H, s, *gem*-CH_3_), 1.53 (1H, t, *J* 12, 2-H_ax_), 1.69 (1H, ddd, *J* 2, 3.5, 12, 2-H_eq_), 1.76 (3H, br s, 5-CH_3_), 1.89 (3H, br s, 13′-CH_3_), 2.04 (3H, br s, 13-CH_3_), 2.11 (1H, br dd, *J* 9.5, 17, 4-H_ax_), 2.28 (1H, br dd, *J* 5.5, 17, 4-H_eq_), 3.97 (1H, m, 3-H), 4.42 (2H, s, CH_2_O), 6.21 (1H, br d, *J* 16, 7-H), 6.32 (1H, br d, *J* 12, 14-H), 6.35 (1H, br d, *J* 12, 10-H), 6.41 (1H, d, *J* 15, 12-H), 6.50 (1H, d, *J* 16, 8-H), 6.69 (1H, dd, *J* 12, 14.5, 15′-H), 6.87 (1H, dd, *J* 12, 15, 11-H), 6.96 (1H, br d, *J* 12, 14′-H), 7.03 (1H, dd, *J* 11.5, 14, 15-H), 9.46 (1H, s, CHO); δ_C_ (125 MHz) 4.50 (C × 3), 4.93 (C × 3), 6.83 (C × 3), 6.87 (C × 3), 9.58, 13.10, 21.70, 28.62, 30.22, 37.04, 43.13, 48.78, 63.81, 65.27, 126.18, 126.99, 127.36, 127.50, 127.95, 128.18, 131.26, 136.97, 137.41, 137.60, 137.81, 138.31, 141.46, 148.81, 194.43; HRMS (ESI) *m/z* calcd for C_37_H_63_O_3_Si_2_ [M+H]^+^ 611.4310, found 611.4320.

### 3.3. Synthesis of C_15_-Phosphonate ***25***

(3*R*,6*R*)-3-Hydroxy C_15_-aldehyde **20**. According to the Khachik’s method, racemic 3-hydroxy C_15_-aldehyde **20** was synthesized starting from commercially available α-ionone and its enantiomers were directly separated using a semi-preparative chiral HPLC column [CHIRALPAK IF 2.0 × 25 cm (Daicel, Tokyo, Japan)] as shown in Figure 2b. Spectral data of the resulting enantiomers were identical with those reported [13].

(1*R*,4*R*)-4-[(1*E*,3*E*)-5-hydroxy-3-methylpenta-1,3-dien-1-yl]-3,5,5-trimethylcyclohex-2-en-1-yl acetate (**23**). Ac_2_O (1.22 mL, 12.9 mmol) was added dropwise to a stirred solution of the (3*R*,6*R*)-aldehyde **20** (1.01 g, 4.3 mmol) in dry CH_2_Cl_2_ (18 mL), Et_3_N (3 mL, 21.5 mmol) and DMAP (0.05 g, 0.43 mmol) at rt for 15 min. The resulting mixture was poured into saturated aq. NaHCO_3_ and extracted with AcOEt and washed with brine. The organic layer was dried and evaporated to give the crude aldehyde, which was dissolved in MeOH (16 mL) and NaBH_4_ (0.165 g) was added to it at 0 °C. After being stirred at 0 °C for 10 min, the reaction was quenched by addition of saturated aq. NH_4_Cl. The resulting mixture was evaporated and the mixture was extracted with AcOEt and washed with brine, dried and evaporated to give a residue, which was purified flash CC (acetone-hexane, 3:7) to provide the allylic alcohol **23** (1.05 g, 87 % from **20**) as a pale yellow viscous oil: [α]^24^_D_ +335.4 (*c* 1.08, CHCl_3_); ν_max_/cm^−1^ 3610 and 3447 (OH), 1720 (CO), 1623 and 1645 (C=C); δ_H_ (300 MHz) 0.87 and 0.99 (each 3H, s, *gem*-CH_3_), 1.45 (1H, dd, *J* 5.5, 14, 2-H), 1.64 (3H, br s, 5-CH_3_), 1.78 (3H, br s, 9-CH_3_), 1.83 (1H, dd, *J* 5.5, 14, 2-H), 2.04 (3H, s, CH_3_COO), 2.38 (1H, br d, *J* 9.5, 6-H), 4.28 (2H, d, *J* 7, 11-H_2_), 5.32 (1H, m, 3-H), 5.41 (1H, dd, *J* 9.5, 15.5, 7-H), 5.49 (1H, m, 4-H), 5.62 (1H, br t, *J* 7, 10-H), 6.08 (1H, d, *J* 15.5, 8-H); δ_C_ (75 MHz) 12.71, 21.41, 22.85, 25.13, 28.86, 33.23, 39.45, 54.55, 59.19, 68.80, 119.98, 128.54, 128.77, 135.90, 136.75, 140.21, 170.92; HRMS (ESI) *m/z* calcd for C_17_H_26_O_3_Na [M + Na]^+^ 301.1774, found 301.1774.

(1*R*,4*R*)-4-[(1*E*,3*E*)-5-diethoxyphosphoryl-3-methylpenta-1,3-dien-1-yl]-3,5,5-trimethylcyclohex-2-en-1-yl acetate (**24**). To a stirred suspension of ZnI_2_ (1.69 g, 5.29 mmol) in dry THF (2.6 mL) was added dropwise P(OEt)_3_ (1.84 mL, 10.6 mmol) and a solution of the alcohol **23** (0.98 g, 3.52 mmol) in dry THF (5.0 mL) at rt. After being stirred at 85 °C for 30 min, the resulting mixture was diluted with H_2_O and AcOEt. The resulting mixture was filtered through a pad of Celite and the resulting mixture was poured into saturated aq. NaHCO_3_ and extracted with AcOEt. The extracts were washed with brine, dried and evaporated to give a residue, which was purified by flash CC (AcOEt-hexane, 3:7) to provide phosphonate **24** (1.13 g, 81%) as a colorless oil: [α]^24^_D_ +251.1 (*c* 0.93, CHCl_3_); ν_max_/cm^−1^ 1720 (CO); δ_H_ (300 MHz) 0.86 and 0.99 (each 3H, s, *gem*-CH_3_), 1.31 (6H, t, *J* 7, OCH_2_C*H*_3_ × 2), 1.45 (1H, dd, *J* 5.5, 13.5, 2-H), 1.63 (3H, br s, 5-CH_3_), 1.77 (3H, dd, *J* 1, 4, 9-CH_3_), 1.83 (1H, dd, *J* 6, 13.5, 2-H), 2.04 (3H, s, CH_3_COO), 2.36 (1H, br d, *J* 9.5, 6-H), 2.76 (2H, dd, *J* 8, 23, CH_2_P), 4.04–4.17 (4H, m, OCH_2_ × 2), 5.31 (1H, m, 3-H), 5.35 (1H, ddd, *J* 2, 9.5, 15.5, 7-H), 6.09 (1H, dd, *J* 1, 15.5, 8-H); δ_C_ (75 MHz) 12.71 (*J*_cp_ 2.3), 16.36 (*J*_cp_ 6.3, C × 2), 21.38, 22.83, 25.12, 26.74 (*J*_cp_ 139.4), 28.85, 33.19 (*J*_cp_ 1.7), 39.43, 54.51, 61.85 (*J*_cp_ 6.8, C × 2), 68.75, 118.40 (*J*_cp_ 11.9), 119.93, 127.53 (*J*_cp_ 4.0), 136.65 (*J*_cp_ 5.1), 137.38 (*J*_cp_ 14.9), 140.24, 170.81; HRMS (ESI) *m/z* calcd for C_21_H_35_O_5_NaP [M + Na]^+^ 421.2114, found 421.2114.

Diethyl {(2*E*,4*E*)-3-methyl-5-[(1*R*,4*R*)-2,6,6-trimethyl-4-triethylsilyloxycyclohex-2-en-1-yl]penta-2,4-dien-1-yl}phosphonate (**25**). To a solution of phosphonate **24** (0.83 g, 2.09 mmol) was added MeOH (25 mL), and NaOMe (28% NaOMe, 1.6 mL, 8.36 mmol) at rt and the mixture was stirred at rt for 30 min. The resulting mixture was poured into saturated aq. NH_4_Cl and extracted with AcOEt. The extracts were washed with brine, dried and evaporated to give the crude alcohol. After TESCl (0.45 mL, 2.68 mmol) was added dropwise to a stirred solution of the crude alcohol (0.796 g, 2.23 mmol) in dry CH_2_Cl_2_ (9 mL), Et_3_N (0.93 mL, 6.69 mmol) and DMAP (13.6 mg, 0.11 mmol) at 0 °C. After being stirred at 0 °C for 15 min. The resulting mixture was evaporated and the mixture was extracted with AcOEt and washed with brine, dried and evaporated to give a residue, which was purified flash CC (acetone-hexane, 3:7) to provide TES-protected phosphonate **25** (0.915 g, 93% from **24**) as a colorless oil: [α]^24^_D_ +137.5 (*c* 1.00, CHCl_3_); δ_H_ (300 MHz) 0.62 (6H, q, *J* 7.5, SiC*H*_2_CH_3_ × 3), 0.82 and 0.94 (each 3H, s, *gem*-CH_3_), 0.98 (9H, t, *J* 7.5, SiCH_2_C*H*_3_ × 3), 1.31 (6H, t, *J* 7 Hz, OCH_2_C*H*_3_ × 2), 1.39 (1H, dd, *J* 8, 13, 2-H), 1.57 (3H, br s, 5-CH_3_), 1.70 (1H, dd, *J* 6, 13, 2-H), 1.77 (3H, br d, *J* 4, 9-CH_3_), 2.38 (1H, br d, *J* 10, 6-H), 2.70 (2H, dd, *J* 8, 23, CH_2_P), 4.04–4.16 (4H, m, OCH_2_ × 2), 4.25 (1H, m, 3-H), 5.35 (1H, ddd, *J* 2, 10, 15.5, 7-H), 5.43 (1H, br s, 4-H), 6.08 (1H, d, *J* 15.5, 8-H); δ_C_ (75 MHz) 4.87 (C × 3), 6.84 (C × 3), 12.75 (*J*_cp_ 2.3), 16.39 (*J*_cp_ 6.2, C × 2), 22.76, 23.36, 26.77 (*J*_cp_ 138.9), 29.53, 34.16 (*J*_cp_ 1.1), 45.66, 54.46, 61.86 (*J*_cp_ 6.8, C × 2), 66.05, 117.99 (*J*_cp_ 12.0), 125.69, 128.27 (*J*_cp_ 4.0), 136.48, 136.73 (*J*_cp_ 5.1), 137.49 (*J*_cp_ 14.3); HRMS (ESI) *m/z* calcd for C_25_H_47_O_4_NaPSi [M + Na]^+^ 493.2874, found 493.2865. 

### 3.4. Synthesis of Loroxanthin (***1***)

To a solution of the TES-protected apocarotenal **8** (245 mg, 0.40 mmol) and the TES-protected phosphonate **25** (226 mg, 0.48 mmol) in dry THF (15 mL) were added NaN(TMS)_2_ (1.0 M in THF; 0.96 mL, 0.96 mmol) at −40 °C. After being stirred at −40 °C for 30 min, the mixture was poured into saturated aq. NH_4_Cl and extracted with AcOEt. The organic layer was washed with brine, dried and evaporated to give the residue was purified by flash CC (AcOEt-hexane, 8:92) to give the condensed product (247 mg, 67%) as a red viscous oil. This was dissolved in dry THF (10 mL) and TBAF (1.0 M in THF; 1.04 mL, 1.04 mmol) was added to it at rt. After being stirred at rt for 15 min, the mixture was poured into saturated aq. NH_4_Cl and extracted with AcOEt. The organic layer was washed with brine, dried and evaporated to give the residue was purified by flash CC (MeOH-CH_2_Cl_2_-acetone, 2:85:15 to 3:80:15) to give loroxanthin (**1**) (143 mg, 61% from apocarotenal **8**) as red solids: λ_max_(EtOH)/nm 268, 423sh, 447, 476; δ_H_ (500 MHz) 0.85 and 1.00 (6H, s, 1′-*gem*-CH_3_), 1.08 (6H, s, 1-*gem*-CH_3_), 1.37 (1H, dd, *J* 6.5, 13, 2′-H), 1.48 (1H, t, *J* 12, 2-H_ax_), 1.63 (3H, br s, 5′-CH_3_), 1.75 (3H, br s, 5-CH_3_),1.77 (1H, overlapped, 2-H_eq_), 1.84 (1H, dd, *J* 6, 13, 2′-H), 1.91 (3H, br s, 9′-CH_3_), 1.97 (6H, br s, 13-CH_3_ and 13′-CH_3_), 2.05 (1H, br dd, *J* 9, 17, 4-H_ax_), 2.39 (1H, br dd, *J* 5.5, 17, 4-H_eq_), 2.41 (1H, br d, *J* 10, 6′-H), 4.00 (1H, m, 3-H), 4.25 (1H, m, 3′-H), 4.55 (2H, br s, 9-CH_2_), 5.44 (1H, dd, *J* 10, 15, 7′-H), 5.55 (1H, br s, 4′-H), 6.05 (1H, d, *J* 16, 8-H), 6.14 (1H, br d, *J* 12, 10′-H), 6.14 (1H, d, *J* 15, 8′-H), 6.24 (1H, d, *J* 11.5, 10-H), 6.26 (1H, br d, *J* 10, 14′-H), 6.29 (1H, br d, *J* 10, 14-H), 6.34 (1H, br d, *J* 16, 7-H), 6.36 (1H, d, *J* 14.5, 12′-H), 6.43 (1H, d, *J* 14.5, 12-H), 6.62 (1H, dd, *J* 12, 14.5, 11′-H), 6.61–6.66 (2H, m, 15-H and 15′-H), 6.71 (1H, dd, *J* 11.5, 14.5, 11-H); δ_C_ (125 MHz) 12.82 (13-CH_3_ and 13′-CH_3_), 13.10 (9′-CH_3_), 21.66 (5-CH_3_), 22.85 (5′-CH_3_), 24.25 (1′-CH_3_), 28.74 (1-CH_3_), 29.49 (1′-CH_3_), 30.29 (1-CH_3_), 34.02 (C1′), 37.13 (C1), 42.54 (C4), 44.63 (C2′), 48.39 (C2), 54.95 (C6′), 57.40 (9-CH_2_), 65.02 (C3), 65.90 (C3′), 123.47 (C11), 124.50 (C4′), 125.07 (C11′), 126.52 (C7 or C12′), 126.62 (C5), 128.85 (C7′), 129.83 (C15 or C15′), 130,74 and 130.76 (C10′ and C15 or C15′), 132.41 (C14′), 133.31 (C10), 133.76 (C14), 135.26 (C9′), 135.46 (C8), 136.04 (C13), 136.90 (C13′), 137.24 (C9), 137.46 (C7 or C12′), 137.65 (C6), 137.69 (C8′), 137.93 (C5′), 139.86 (C12); HRMS (APCI) *m/z* calcd for C_40_H_55_O_3_ [M − H]^−^ 583.4157, found 583.4151.

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
