# Peer review of "Total Synthesis of Loroxanthin"

_marinedrugs, 2022, doi:10.3390/md20110658_

Round 1
Reviewer 1 Report
The submitted manuscript entitled “Total Synthesis of Loroxanthin” by Prof. Yamano described an efficient and concise total synthesis of carotenoid loroxanthin. The synthetic route utilized a Wittig reaction, a Stille coupling reaction, and an Horner–Wadsworth–Emmons reaction to couple some simple fragments, and provided sufficient material for further biological tests of this natural product. The manuscript is written in a clear and engaging fashion and is relatively free of errors. The references and supplementary materials are appropriate and the compounds are well-characterized. Thus, I recommend publication of this manuscript in Marine Drugs following some cleaning up, which are listed below:
1. Page 4, scheme 4: "d, e" should be "d", "f, g" should be "e, f".
2. Page 10: loroxanthin (1) is not fully characterized – melting point and optical rotation data are missing.
3. Page 11, ref 17: "FEBS Lett." should be in italic format.
Author Response
Thank you for reviewing our manuscript.
- Alcoholysis conditions of compound 24 were missing. So, I corrected the condition description in Scheme 4.
- I am reluctant to measure the optical rotation power. In the field of carotenoids, CD spectra have been widely used instead of optical rotation powers. I am also reluctant to measure the melting point of 1. Because 1 is too thermally unstable to carry out crystallization that is necessary to measure reliable melting points. For this reason, generally melting points of synthesized or isolated carotenoids have been rarely reported.
- I corrected it as you suggested.
Reviewer 2 Report
Page 4; Line 91-100 : Did you attempt to prepare compound 22 from 21 using Mayer’s method?
Page 4, line Line 98: allylic alcohol 23, make sure it is not allylic alcohol 21. If you did not attempt to prepare compound 22 from allylic alcohol 21 so you can skip the details of transformation of 21 to 22 that have been done by Mayer and co-workers.If you try to prepare 22 from 23 so you should give the details what you have tried.
Conclusion : Summarize the total yield and total steps in the longest linear sequence.
Page 6; line 161 : Chemical shift.
Page 6; line 162 : Coupling constant (J)
Page 6; line 197: 3.53 and 3.68 (each 1H, qd, 7.2, 9.3, OCH2CH3) should be 3.53 (qd, J= 7.2, 9.3 Hz, 1H, OCH2CH3), 3.68 (qd, J= 7.2, 9.3 Hz, 1H, OCH2CH3)……do the same with other.
Check report format for NMR with MDPI. If no format, I recommend the author use the following format for 1H-NMR
dH (300 MHz, CDCl3) 0.08 (s, 6H, SiCH3x2), 0.89 (s, 9H, tert-Bu), 2.52 (t, 1H, J= 6.5 Hz, OH),……..
Page 10: 13C of compound 25 missing.
Supporting information
1H-NMR : spectra should be in the same scale, may be from -1 ppm to 12 ppm. All the 1H-NMR peaks should show integration number and chemical shift.
Author Response
# 1 We haven’t attempted the preparation of compound 22 from 21 by considering the assumption that the conversion of allyl alcohols 21 and 23 to 22 would be proceed via an identical intermediate.
In precedent study, various carotenoids have been prepared from allyl alcohols including 21 and 23 via C15-phosphonium salts. Difficulties on preparation of the related phosphonium salts would be worth disclosing for researchers working in this field. I dare to report on the manuscript as it is, including successes and failures.
# 2 According to the suggestion, total yield and total steps were described in the text (page 4, line 118).
#3 I corrected the spell of chemical shifts (page 6 line 161) and J to italic (page 6 line 162).
# 4 I added 13C NMR data of compound 25.
#5 Unfortunately, I couldn’t find MDPI’s recommendation for NMR data format and supporting information. I am hoping to keep the current format for NMR data that I use in manuscripts of this journal and other journals to make consistency in my disclosing data.
Reviewer 3 Report
This manuscript describes the total synthesis of Loroxanthin, an in-chain hydroxylated carotenoid. The authors synthesized this natural product by the condensation, via a HWE reaction, of subunits 8 and A.
The authors start by describing the synthesis of subunit 8, which is prepared from three building blocks (compounds 9, 10 and 12). The synthesis of two of these compounds (9 and 12) have already been described and the synthesis of the third one (compound 10) is simple and has no synthetic interest. Moreover, the last step of this synthesis furnishes a mixture of isomers. This makes that the efficiency of the synthetic route is very low.
Concerning subunit A, its preparation requires the HPLC separation of a synthetically advanced racemic starting material whose synthesis has already been described in the literature. The complexion of the synthesis of this fragment involves classical reactions.
The SI of this manuscript is of high quality.
In conclusion, the synthesis of most of the building blocks used to prepare Loroxanthin have already been described and the reactions used to join these fragments do not present any synthetic interest being the significance of this work low.
For the above reasons, I regret that I could not recommend this manuscript for publication in Marine Drugs.
Author Response
Just looking at the successful reaction pathway from classical reactions, the reader may not find them of synthetic interest.
However, behind the successes, there are various attempts to take into account the chemical specificity of the carotenoid components, and we believe that disclosing them would be of great benefit to researchers in this field.
We also believe that the accumulation of knowledge on synthetic methods is essential to enhance research on naturally occurring carotenoids, which hold great promise for biological activity.